# Use of Oxidative Stress Responses to Determine the Efficacy of Inactivation Treatments on *Cryptosporidium* Oocysts

**DOI:** 10.3390/microorganisms9071463

**Published:** 2021-07-08

**Authors:** Tamirat Tefera Temesgen, Kristoffer Relling Tysnes, Lucy Jane Robertson

**Affiliations:** 1Laboratory of Parasitology, Department of Paraclinical Sciences, Faculty of Veterinary Medicine, Norwegian University of Life Sciences, Oluf Thesens vei 22, 1433 Ås, Norway; kristoffer.tysnes@nmbu.no (K.R.T.); lucy.robertson@nmbu.no (L.J.R.); 2NABAS AS, Moer Allé 33, 1435 Ås, Norway

**Keywords:** *Cryptosporidium*, gene expression, oocyst, viability, RNA-Seq, RT-qPCR

## Abstract

*Cryptosporidium* oocysts are known for being very robust, and their prolonged survival in the environment has resulted in outbreaks of cryptosporidiosis associated with the consumption of contaminated water or food. Although inactivation methods used for drinking water treatment, such as UV irradiation, can inactivate *Cryptosporidium* oocysts, they are not necessarily suitable for use with other environmental matrices, such as food. In order to identify alternative ways to inactivate *Cryptosporidium* oocysts, improved methods for viability assessment are needed. Here we describe a proof of concept for a novel approach for determining how effective inactivation treatments are at killing pathogens, such as the parasite *Cryptosporidium.* RNA sequencing was used to identify potential up-regulated target genes induced by oxidative stress, and a reverse transcription quantitative PCR (RT-qPCR) protocol was developed to assess their up-regulation following exposure to different induction treatments. Accordingly, RT-qPCR protocols targeting thioredoxin and *Cryptosporidium* oocyst wall protein 7 (COWP7) genes were evaluated on mixtures of viable and inactivated oocysts, and on oocysts subjected to various potential inactivation treatments such as freezing and chlorination. The results from the present proof-of-concept experiments indicate that this could be a useful tool in efforts towards assessing potential technologies for inactivating *Cryptosporidium* in different environmental matrices. Furthermore, this approach could also be used for similar investigations with other pathogens.

## 1. Introduction

Environmentally transmitted pathogens represent a major public health concern worldwide. These pathogens, which can usually survive for prolonged periods in the external environment, are often transmitted to humans via contaminated environmental matrices, such as water and food [1]. Although methods for inactivating waterborne pathogens (e.g., UV irradiation) have been established, these are not always applicable to other matrices including various sorts of foods. Therefore, different technologies such as ozonation, high hydrostatic pressure, pulsed light, and thermal methods have been explored to inactivate microbial contaminants on environmental matrices such as fresh produce [2,3,4,5]. Such investigations require reliable methods for assessing the efficacy of inactivation technologies, especially for pathogens that are non-culturable or difficult to culture. Among such pathogens are environmentally transmitted protozoan parasites such as *Cryptosporidium*, *Cyclospora*, *Toxoplasma*, etc.; the present study focused on *Cryptosporidium*.

*Cryptosporidium* is a coccidian parasite responsible for cryptosporidiosis, a gastrointestinal disease manifesting mainly as watery diarrhea, nausea, vomiting, fatigue, and other signs and symptoms [6]. The watery diarrhea typical for cryptosporidiosis can sometimes be profuse and prolonged and result in dehydration and wasting. This may cause critical illness [7] and, due to the paucity of effective treatments, mortality rates may be high, especially for immunocompromised patients. Some species of *Cryptosporidium*, such as *Cryptosporidium parvum*, are zoonotic, with a wide range of potential hosts, although young ruminants are predominantly associated with human infections.

Humans acquire the infection through ingestion of the sporulated oocysts, each of which contain four sporozoites that invade the cells of intestinal epithelium. The highly robust and environmentally resistant oocysts are well suited for transmission via contaminated environmental matrices, such as water and food. Thus, transmission is often due to ingestion of water or food that has been contaminated with infectious oocysts. Outbreaks of cryptosporidiosis attributed to swimming pool or drinking water contamination have been frequently recorded [8], and foodborne outbreaks have also been documented [9,10]

Due to the complex nature of parasites, it has been challenging to develop sensitive in vitro methods for assessing their viability. The current “gold standard” method for viability testing of parasites is in vivo animal bioassay. Alternative in vitro techniques that have been explored include experimental infections in cell cultures, in vitro excystation, sporulation, vital dye inclusion/exclusion, propidium monoazide PCR (PMA-PCR), nucleic acid sequence-based amplification (NASBA), and reverse transcription quantitative PCR (RT-qPCR) [11], with each method having advantages and disadvantages. In vitro cell culture has been widely used for assessing the efficacy of *Cryptosporidium* inactivation by chlorination and UV irradiation among others [12,13,14,15]. It has been shown that the in vitro excystation method overestimated the viability of *Cryptosporidium* oocysts when, in particular, assessing the efficacy of UV inactivation, which was thus suggested to be less efficacious [16].

Although there are several published protocols for viability assessment of *C. parvum* based on RT-qPCR, there is no consensus on the best approach and there are different rationales for selecting a genetic marker. The present study aimed at developing a comparative RT-qPCR method that could be used for assessing the efficacy of inactivation technologies on pathogens, such as protozoan cysts and oocysts or robust bacterial or fungal spores, using *C. parvum* as a model organism. Potential target genes were identified by their up-regulation in response to oxidative stress.

## 2. Materials and Methods

### 2.1. Parasites

*Cryptosporidium parvum* IOWA strain oocysts were purchased from Bunch Grass Farm (Deary, ID, USA) and used within three months. The oocysts had been stored in PBS supplemented with antibiotics (1000 IU penicillin, 1000 μg streptomycin), wrapped in cold gel pack, and shipped overnight by courier. An additional isolate of *C. parvum* oocysts was purified from calf feces that had been submitted to the Parasitology lab, Norwegian University of Life Sciences for diagnostic examination. The comparison of glycoprotein 60 (GP60) gene fragments from this isolate with the IOWA strain showed about 87% identity. The CryptoGenotyper tool [17] classified the IOWA strain as subtype IIaA17G2R1, whereas the isolate from the Parasitology lab was subtype IId.

### 2.2. Experimental Setup

In order to achieve the objective of the study, the experimental setup was designed to include four main sections including: (i) exploration of different induction approaches to stimulate upregulation of a range of genes; (ii) identification of these genes using RNA-Seq and differentially expressed gene (DEG) analysis; (iii) selection of appropriate targets and designing RT-qPCR protocols for their relative quantification; (iv) use of these protocols to determine viability of oocysts following exposure to different chemical and physical potential inactivation treatments. The experimental setup is summarized in Figure 1.

#### 2.2.1. Identification of Inducible Target Genes Using RNA-Seq and DEG Analysis

##### Induction of Upregulation of Gene Expression

Five different gene-expression induction approaches were explored using, in brief: (a) two concentrations of menadione sodium bisulfite (MSB); (b) an enzymatic reaction composed of xanthine oxidase and hypoxanthine, and (c) heat shock at two different temperatures; see Table 1 for details. All chemicals used were purchased from Sigma-Aldrich, Norway. Each of the 5 induction groups contained four independent replicates of approximately 10 million *C. parvum* oocysts. Oocysts were pre-washed twice in water before being subject to the different induction treatments. A control group was stored refrigerated prior to RNA extraction.

##### RNA Extraction and Quality Assessment for RNA-Seq

The RNA extraction protocol for this study was based on RNeasy Plus Mini Kit (Qiagen, Oslo, Norway), with slight modifications to the lysis approach. Briefly, oocyst lysis was performed in a lysing matrix E tube (MP Biomedicals, Illkirch Cedex, France) to which 600 µL of the RLT plus buffer was added, and the tube subjected to 2 cycles of bead-beating at 4 m/s for 25 s with a 3-min pause on ice between cycles. The lysate was centrifuged at 12,000× *g* for 2 min and then added to the gDNA eliminator spin column before elution into 55 µL of nuclease-free water and storage at −20 °C.

RNA quality was assessed using an Agilent 2100 Bioanalyzer and the RNA 6000 Nano kit was used for sample preparation. The RNA integrity number (RIN) produced by the Bioanalyzer provides an indication of the RNA quality; RIN ranges from 1–10, where 10 indicates intact RNA. The RNA extracted from samples subjected to heat shock at 45 °C for 20 min did not produce RIN values and hence was not submitted for RNA-Seq.

##### RNA Sequencing (RNA-Seq)

The RNA samples were sequenced at a national technology core facility (Norwegian Sequencing Centre, Ullevål Hospital, Oslo, Norway). The RNA library preparation was performed using Truseq stranded RNA prep kit and sequenced in the NovaSeq SP flow cell with the NovaSeq 6000 sequencer. After indexing, all samples were sequenced in the same flow cell using single-read sequencing of 100 bp and a sequencing depth of ca 20 million reads per sample.

#### 2.2.2. RT-qPCR Method Development

The gene expression induction approach selected for the RT-qPCR method development was based on the oxidative stressor challenge as shown in Figure 2.

##### Target Genes and Primers

Based on the differentially expressed genes (DEG) analysis (see Section 2.3.2), six different target genes were selected for further testing using RT-qPCR. These genes included COWP7, type 3 malate dehydrogenase, thioredoxin, prohibitin, heat shock protein 70 (HSP70), and UDP-glucose 6-dehydrogenase (UGDH). In order to evaluate the targets identified by the DEG analysis by RT-qPCR, appropriate primers were designed in Geneious Prime, with intron-spanning sites included where possible (Table 2). In addition to the differentially expressed target genes, 18S rRNA was selected as the reference gene to normalize variation between samples.

##### RNA Extraction for the RT-qPCR Method

The PureLink RNA mini kit (Thermo Fisher Scientific, Oslo, Norway) was employed for total RNA extraction in the RT-qPCR method development. This kit was used here for pragmatic reasons; it was available in the lab and had demonstrated no difference compared with the RNeasy Plus Mini Kit (Appendix A). Briefly, the sample containing approximately 2 million oocysts of *C. parvum* (0.3 mL) was mixed with 0.6 mL of the lysis buffer containing 40 mM dithiothreitol in a lysing matrix E tube and subjected to bead-beating (two cycles of 4 m/s for 25 s with 3 min pause on ice in between the cycles). The lysate was then centrifuged at 12,000× *g* for 2 min and the supernatant transferred to a new collection tube and purified according to the kit’s instruction. RNA was eluted in 50 µL RNase free water before DNase treatment using Turbo DNA free kit (Thermo Fisher Scientific, Norway) and stored at −20 °C until RT-qPCR analysis.

##### Reverse Transcription Quantitative PCR (RT-qPCR)

One-Step SYBR Green RT-qPCR

RT-qPCR was performed in a total volume of 20 µL, which included 10 µL of One-Step SYBR Green master mix, 0.4 µL of the qScript RT-mix, 0.5 µM of each primer, and 2 µL of RNA template. The reaction assembly was performed according to the instructions provided with the master mix (Quantabio, Beverly, CA, USA) and thermal cycling and fluorescence measurement were done in Stratagene Mx3005P qPCR instrument (Agilent Technologies, Matriks As, Oslo, Norway). The thermal profile of the reaction was as follows: cDNA synthesis at 50 °C for 10 min, initial denaturation at 95 °C for 5 min followed by 45 cycles of denaturation at 95 °C for 15 s, and annealing/extension at 60 °C for 1 min. Additionally, a melt-curve analysis step was included, with a gradual increase of temperature from 65 °C to 95 °C and fluorescence data collected every 0.1 °C.

One-Step Probe RT-qPCR

In addition to the SYBR Green RT-qPCR, a probe-based One-step RT-qPCR was tested for the COWP7 target. The reaction was performed in a total volume of 20 µL that contained 10 µL of KiCqStart^®^ One-Step Probe RT-qPCR ReadyMix, 0.5 µM of each primer, 0.25 µM probe, and 2 µL of the RNA template. The 18S rRNA target was detected and quantified using 0.6 µM of the forward primer (1:1 combination of JF1 and JF2) and reverse primer (JR) and 80 nM of the probe (JT2). Thermal cycling and fluorescence measurement were done in Stratagene Mx3005P qPCR instrument (Agilent Technologies). The thermal profile of the reaction included cDNA synthesis at 50 °C for 10 min, initial denaturation at 95 °C for 3 min followed by 45 cycles of denaturation at 95 °C for 15 s, annealing at 60 °C for 1 min, and extension at 72 °C for 30 s.

Standard curves were prepared to evaluate the linearity and efficiency of each primer pair. An additional test was conducted to compare xanthine and hypoxanthine as the substrate in the oxidative stress induction.

#### 2.2.3. Evaluation of the RT-qPCR Method

The RT-qPCR method was evaluated for its applicability in the differentiation of viable and inactivated oocysts using different inactivation treatments. The main steps of the RT-qPCR are summarized in the flow chart presented in Figure 2.

##### 2.2.3.1. Inactivation of *Cryptosporidium* Oocysts

In order to assess the applicability of the RT-qPCR method, physical and chemical inactivation approaches that had been previously evaluated using mice bioassay [19,20], were used to determine whether the method developed here could be used to evaluate inactivation efficacy. Accordingly, *Cryptosporidium* oocysts were inactivated by heating at 80 °C (on heat block) for 3 min and incubated at room temperature for 3 h prior to RNA extraction. This was considered sufficient for complete inactivation of the oocysts and therefore used to prepare different proportions of live/inactivated oocysts as described in Section 2.2.3.2. Other inactivation treatments investigated in this study included heating at 60 °C for 2 min, freezing at −20 °C for 2 h, 24 h, and 48 h. In addition to thermal inactivation, chemical treatment was assessed by subjecting oocysts to 4 mg/L and 0.2 mg/L free chlorine concentrations for 30 min.

##### 2.2.3.2. Distinguishing between Viable and Inactivated Oocysts Using the New RT-qPCR Method

The new RT-qPCR method was evaluated for its ability to discriminate between different proportions of viable and inactivated oocysts by analyzing samples containing different proportions of viable and inactivated oocysts. In brief, the oocyst mixtures (containing viable/inactivated oocysts (heated at 80 °C for 3 min) in the following proportions: 0/100, 1/99, 10/90, 100/0%) were exposed to an inactivation regime as described in Section 2.2.3.1. Then the oocysts were exposed to xanthine oxidase catalyzed oxidative stressor before RNA extraction. RT-qPCR targeting thioredoxin, COWP7, and 18S rRNA was used to determine relative quantity of thioredoxin and COWP7 genes from exposure to the oxidative stressor. Controls that contained the same proportion of viable/inactivated oocysts, but had not been exposed to the oxidative stressor, were included in each run.

### 2.3. Statistical Analysis

#### 2.3.1. Data Pre-Processing and Mapping

Raw sequence data (fastq files) from the RNAseq were trimmed for adapter sequences and low quality reads (having phred score <33) by using trimommatic 0.39 (http://www.usadellab.org/cms/?page=trimmomatic, accessed on 26 February 2021) and assessed for quality by using FASTQC tool. The trimmed reads were mapped to the reference genome of *C. parvum* IOWA II obtained from CryptoDB release 46 (https://cryptodb.org/common/downloads/release-46/CparvumIowaII/fasta/data/CryptoDB-46_CparvumIowaII_Genome.fasta, accessed on 26 February 2021) using STAR version 2.5 [21]. In addition, the Salmon tool [22] was applied for quasi-mapping of the trimmed reads against the annotated transcript of *C. parvum* IOWA II available from the CryptoDB release 46 (https://cryptodb.org/common/downloads/release-46/CparvumIowaII/fasta/data/CryptoDB-46_CparvumIowaII_AnnotatedTranscripts.fasta, accessed on 26 February 2021).

#### 2.3.2. Differential Gene Expression Analysis

The mapped reads were counted against the recently updated genome annotation file obtained from CryptoDB release 46 (https://cryptodb.org/common/downloads/release-46/CparvumIowaII/gff/data/CryptoDB-46_CparvumIowaII.gff, accessed 26 on February 2021). Gene counts table was prepared using featureCounts [23], which was then used as an input for the differential expression analysis with the R–based tool, DESeq2 [24].

In addition to the gene-based differential expression analysis, a transcript-based differential expression analysis was performed on the transcript abundance output obtained from the Salmon tool and further analyzed with DESeq2 in R. The differential expression analysis output contained the list of genes with their log2fold change and associated *p*-values. The results of the DEG analysis were visualized using principal component analysis (PCA) plots and MA-plots.

## 3. Results

### 3.1. RNA-Seq Analysis

The sequencing reads were of high quality as indicated by the average Phred score of 36. The RNA-Seq data have been submitted to the sequence read archive of NCBI and are available for public use with the accession number PRJNA669334 (http://www.ncbi.nlm.nih.gov/bioproject/669334, accessed on 7 July 2021). The overall alignment rate of the reads against the reference genome was >96% whereas the alignment rate of the reads against the reference transcriptome was >89% (Appendix A).

Differentially expressed genes (DEG) analysis of the RNA-Seq data revealed that, as shown by the principal component analysis (PCA), two groups of samples were clearly separated from the others that grouped together (Figure 3). The PCA plot shows that the samples subjected to oxidative stressor from the enzymatic reaction of xanthine oxidase and hypoxanthine, as well as those subjected to heat shock, were placed distinctly separate from one another and from the other three groups. The samples treated with menadione sodium bisulfite (MSB) showed no difference from the control samples.

The MA plot (Appendix A) for the RNA-Seq data also showed differences in gene expression between the sample groups, indicating that MSB-treated samples and untreated control samples had similar gene expression levels, as shown by the distribution of genes around the horizontal axis. In contrast, samples subjected to heat shock or to the xanthine oxidase catalyzed reaction had clearly visible differences in gene expression levels compared with the untreated control.

The DEG analysis showed that many genes were differentially expressed in response to the oxidative stress (Dataset S1). In order to develop an RT-qPCR method, six target genes were selected based on their log2fold change and their general biological relevance in the oxidative stress response of cells (Appendix A).

### 3.2. RT-qPCR Method Development

The performance of each primer pair (i.e., efficiency and linearity) was well within the acceptable ranges. The standard curve prepared to assess the efficiency and linearity of the RT-qPCR based on the COWP7 gene is presented as an example (Appendix A).

Preliminary experiments investigating the six different target genes selected for the RT-qPCR method development showed that all were upregulated following oxidative stress, as indicated by consistently lower C_q_ values than those of untreated controls (Table 3). This confirmed the results obtained from the DEG analysis of the RNA-Seq data.

Among the six target genes evaluated, COWP7 and thioredoxin were selected for further evaluation. The intron-spanning primer pair designed for the detection of COWP7 showed no amplification from genomic DNA in the absence of DNase treatment, confirming the reliability of comparative quantitation of the transcript among the different groups of samples. For thioredoxin, the DNase treatment of the RNA sample with Turbo DNA-free kit was effective, as no amplification was detected in the absence of the reverse transcriptase enzyme.

The comparison of xanthine and hypoxanthine as a substrate for the xanthine oxidase catalyzed reaction showed no significant difference. However, samples treated with xanthine had consistently lower C_q_ values when tested with COWP7 and thioredoxin (Appendix A). Therefore, xanthine was employed for the rest of the RT-qPCR evaluation.

Compared with heat shock (45 °C for 20 min), the xanthine oxidase-catalyzed stressor challenge resulted in increased gene expression. Although the heat shock also induced higher gene expression than in controls, it was considerably lower than when oocysts were exposed to the oxidative stress. The difference was about 3 C_q_, which amounts to approximately 8.5-fold (Figure 4).

### 3.3. Evaluation of the RT-qPCR Method

The thioredoxin RT-qPCR results indicated that the level of gene expression was proportionally related to the number of viable oocysts in the sample (Table 4). Induction of the oxidative stress response resulted in significantly more transcripts than that of the respective control that had not been exposed to the oxidative stressor.

The RT-qPCR protocol was further tested on a different isolate of *C. parvum* that had been purified from a sample delivered for diagnosis at the Parasitology lab, Norwegian University of Life Sciences. The results with this isolate indicated that the protocol was applicable to another *Cryptosporidium* isolate, in addition to the IOWA strain used for the development of the method. The results of the comparative RT-qPCR showed that the sample subjected to oxidative stress had log2foldchange of 3 to 3.9.

The method was also evaluated for its use in testing the efficacy of some of the physical and chemical means of inactivation of the oocysts. Although oocysts were completely inactivated following freezing (−20 °C) for 24 h and 48 h, oocysts frozen for only 2 h were apparently not inactivated. This was indicated by the oocysts’ response to the oxidative stressor challenge, with a mean C_q_ (±SD) of 28.5 ± 0.1 for these oocysts. This is very similar to control oocysts that had not been frozen with mean C_q_ (±SD) of 28.6 ± 0.3 (Table 5).

Furthermore, oocysts that had been heated at 60 °C for 2 min showed some, but incomplete, inactivation, compared with the control oocysts. Although the mean C_q_ of these oocysts was lower than that of their corresponding control (not exposed to the oxidative stressor), it was markedly higher than that of the test control oocysts that had not been exposed to the elevated temperature (Table 5). This means that following exposure to this potential thermal-inactivation treatment, some oocysts were still able to respond to the oxidative stressor challenge with gene up-regulation. However, oocysts exposed to 80 °C for 3 min seem to be completely inactivated as the RT-qPCR results showed no amplification for the COWP7 target. However, some traces of amplification could be detected from the thioredoxin target (Table 5).

Chemical treatment of oocysts with 0.2 and 4 mg/L free chlorine for 30 min at room temperature did not affect the viability of the oocysts (Appendix A). The comparative RT-qPCR results show the difference between oocysts treated with bleach and those stored at −20 °C for 24 h.

## 4. Discussion

In this study, we have developed and evaluated a RT-qPCR method that could be useful for differentiating viable, potentially infectious, oocysts of *C. parvum* from inactivated ones. The method development approach was guided by the biological assumption that viable oocysts would respond actively to stressors with up-regulation of specific genes. Thus, as changes in gene expression would not be expected in dead oocysts subjected to exposure to stressors, this could be an indicator that the oocysts were viable. Rigorous testing by RNA-Seq analysis, and with complementary results obtained with the RT-qPCR, indicated this to be the case.

The reactions catalyzed by xanthine oxidase, as shown in (Appendix A), were further explored to compare xanthine and hypoxanthine as the substrate. As oxidation of xanthine produces uric acid, which can also act as a pro-oxidant [25] we assumed that the use of xanthine as substrate might result in greater stimulation of gene expression. Indeed, the results of the comparative experiment showed a stronger response for xanthine than hypoxanthine, although the difference was relatively small. Nevertheless, for the experiments that involved xanthine as substrate, the precision of the quantification was superior.

RNA-Seq analysis revealed several DEGs in response to the oxidative stressor challenge. Although RNA-Seq have been used for transcriptomic analysis of the developmental stages of *Cryptosporidium* [26,27,28], to the best of our knowledge, this technique has not previously been exploited for the purpose of developing methods for viability assessment of *Cryptosporidium* oocysts.

Among the six target genes evaluated, COWP7 and thioredoxin were selected for further evaluation. This was due to the high log fold change for each gene. COWP7 was also selected because of the putative role of COWP genes in protecting the oocysts and surviving environmental stresses [29]. In addition, COWP7 was the only gene among the six examined that contains introns, making it a suitable candidate for RT-qPCR method. Furthermore, thioredoxin was selected on the basis of its potential role in the oxidative stress response, as it is a well-known antioxidant enzyme that protects cells from cytotoxicity elicited by oxidative stress [30].

The most commonly used method for gene-expression induction prior to RNA extraction followed by RT-qPCR for *Cryptosporidium* viability testing is heating the oocysts at 45 °C for 20 min so that the sensitivity would be improved and HSP70 has been the target of choice [20,31,32]. However, in our work, exposure of oocysts to 45 °C for 20 min resulted in poor quality RNA, probably due to degradation, which could not be used for RNA-Seq. Although these samples were not suitable for RNA-Seq, the RT-qPCR results showed that gene expression could be induced by heat shock, but was about 8-fold lower than the present approach of exposure to oxidative stress.

In contrast, another study argued that the ideal viability marker should not be altered by an external stressor and hence suggested the use of CP2 over HSP70 [33]. Others have suggested using the ratio of mRNA to DNA for the assessment of viability following heat-induction [32]. Collectively, these results indicate that RT-qPCR-based methods could be a rapid, sensitive, and reliable estimate of the viability of parasites when selection of target genes, designing the experimental setup, and interpreting the results are optimized.

The approach that we explored involved induction of gene expression by exposure to stressors and comparison of the relative quantity of expression of relevant genes in exposed and non-exposed control samples (normalized against reference genes to correct for artificial differences between the samples). This approach avoids the problem of over-estimation of viability due to detection of residual RNA in inactivated oocysts, as has been previously reported [34,35].

Through thorough evaluation of the RT-qPCR method using oocysts exposed to various inactivating treatments, we have shown that this method may be a useful tool for assessing the efficacy of different treatments at inactivating *Cryptosporidium* oocysts. Furthermore, by using the 18S rRNA gene as a normalizer and inclusion of untreated control samples we have shown that this approach may enable precise estimation of log reductions in oocyst viability due to different inactivation treatments.

The inactivation efficacy of freezing at −20 °C was time dependent, with longer storage duration resulting in more effective inactivation; although oocyst viability was not affected by freezing for 2 h, complete inactivation was seen after 24 h. These findings corroborate the results reported by Fayer and Nerad (1996), in which the mouse infectivity assay was employed to test the effect of low temperatures on the viability of oocysts. According to the results of that study, mice fed with the oocysts stored at −20 °C for 1 h, 3 h, 5 h, and 8 h were infected, as indicated by the histological examination of the colon, ileum, and cecum of the mice showing the parasite’s developmental stages, but mice that received oocysts frozen at −20 °C for 24 h and 168 h were not infected [19].

According to the RT-qPCR method presented here, chemical treatment of oocysts with 0.2 and 4 mg/L free chlorine for 30 min at room temperature did not affect the viability of the oocysts. It has long been known that *Cryptosporidium* oocysts are incredibly resistant to chlorination [36,37]. Thus, although the results are not surprising, they demonstrate and confirm that the new RT-qPCR method could be a useful method for determining the inactivation efficiency of different chemical treatments.

Complete inactivation of the oocysts after heating at 80 °C for 3 min agrees with the findings of Travaillé et al., (2016) [20], in which mice fed with oocysts heated at 80 °C for 2 min were not infected. In addition, Travaillé et al., (2016) reported that mRNA could not be detected when mRNA extraction was performed after incubation of the sample at room temperature for 1 h [20]. However, in contrast, our results were different, as mRNA was detected even after incubation of the heat-treated oocysts at room temperature for 3 h. This could be due to differences in the stability of different types of mRNA, as it has been shown that HSP70 mRNA is more stable than β-tubulin mRNA [34]. Indeed, in the present study, it was shown that thioredoxin was more likely to be detected than COWP7.

The novelty of the present RT-qPCR method lies in the application of an oxidative stressor challenges to assess the metabolic activity among populations of oocysts, as the method exploits the relative quantification of genes expressed in response to oxidative stress; inactivated oocysts will not demonstrate alterations in gene expression. This differs substantially from the assumption used in various other efforts to develop viability assays, where mRNA detection is used as an indicator of viability. However, as we see here, and has also been reported previously [34], mRNA can be detected in inactivated oocysts, and therefore does not adequately differentiate between viable and inactivated oocysts.

However, the present study is not without limitations and further validation should be performed, including comparison with the “gold standard” method (i.e., mouse infectivity bioassay). As we were unable to perform bioassays, the in-house evaluation of the RT-qPCR method was designed to include the parameters of inactivation treatments that had been assessed by bioassay in previously published work [19,20].

Another limitation of the present RT-qPCR method is that the sensitivity of the method is poor. This could be due to the mild lysis protocol used during the RNA extraction. This means that the method is not suitable for assessing the viability of oocysts detected in naturally contaminated water samples, or other environmental matrices, where usually only low numbers of contaminating oocysts are detected. However, further optimization may improve the sensitivity. Nevertheless, the present method’s value will lie in being able to assess the efficacy of potential inactivation procedures, as high numbers of oocysts are used for such investigations. The lysis approach used in the RNA extraction protocol was the same as that used for the RNA-Seq analysis, which is specifically intended to result in high-quality RNA. However, such high-quality RNA might not be necessary for RT-qPCR. Therefore, optimization of the RNA extraction protocol to increase RNA yield, and thus the sensitivity of the RT-qPCR, may be a pertinent approach. However, it is important that the RNA yield is not achieved at the expense of the quality required for the RT-qPCR protocol.

In conclusion, the findings of the present study showed that RNA-Seq and DEG analysis are very useful tools in the identification of targets when developing RT-qPCR based methods for viability testing. It was also shown that oxidative stressors are suitable for inducing mRNA expression of selected genes in *Cryptosporidium* oocysts. Evaluation of the present RT-qPCR method showed that it could be a reliable, rapid, and cost-effective alternative used for testing the efficacy of different inactivation treatments on *Cryptosporidium* oocysts. Moreover, the present approach to method development could also be applied to other environmental pathogens.

## Figures and Tables

**Figure 1 microorganisms-09-01463-f001:**
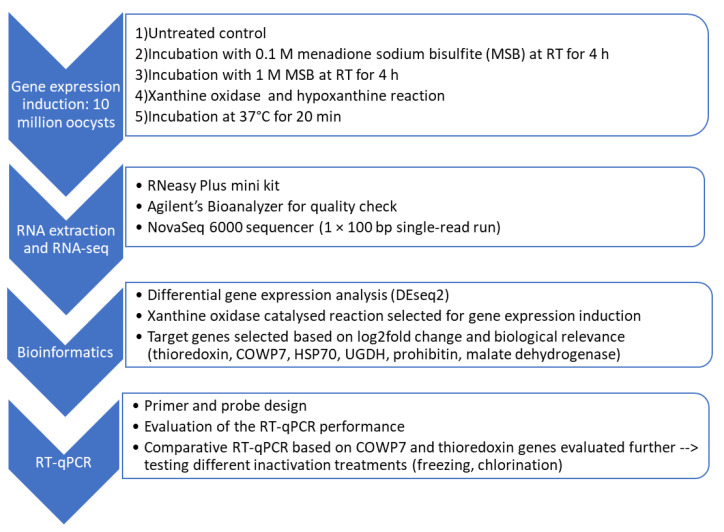
Flow chart indicating the 4 stages of the RT-qPCR method development.

**Figure 2 microorganisms-09-01463-f002:**
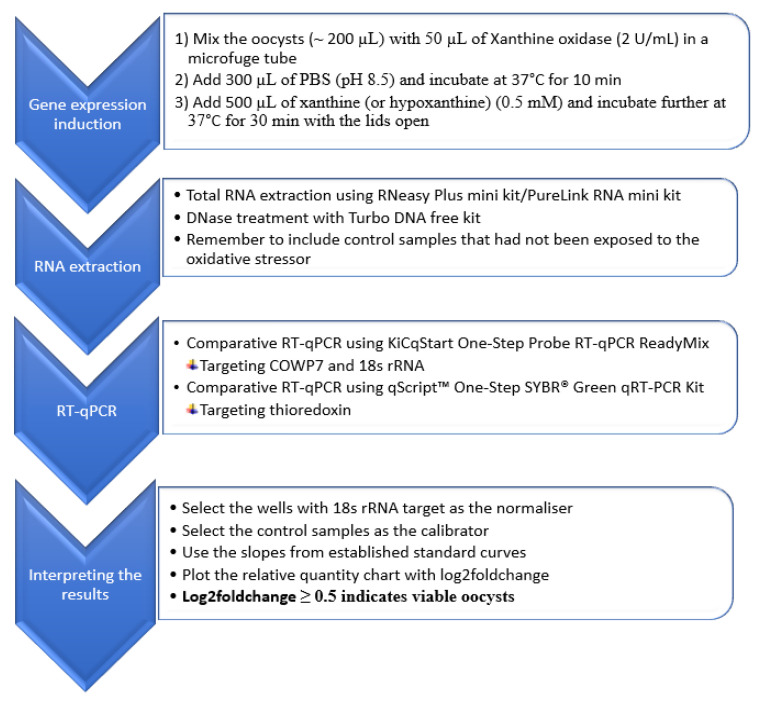
Flow chart representing the summary of the comparative RT-qPCR method for the assessment of viability of *Cryptosporidium* oocysts.

**Figure 3 microorganisms-09-01463-f003:**
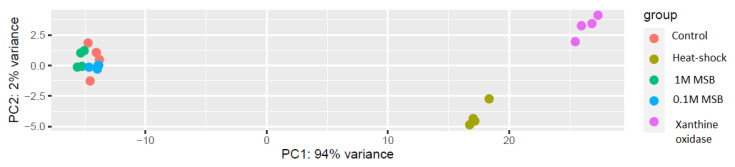
PCA of the RNA-Seq data after rlog transformation.

**Figure 4 microorganisms-09-01463-f004:**
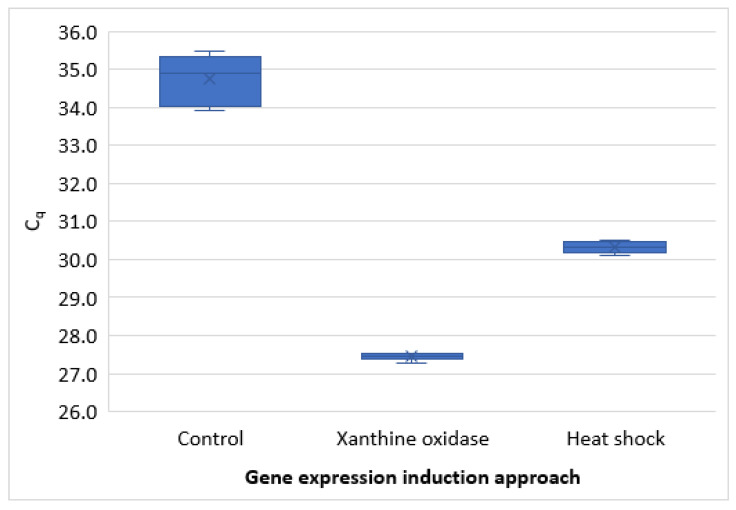
Comparison of gene expression induced by heat shock (45 °C for 20 min) with that induced by oxidative stressor challenge catalyzed by xanthine oxidase.

**Table 1 microorganisms-09-01463-t001:** Description of the gene expression induction approaches explored for RNA-Seq analysis.

Treatment	Brief Description
1 M MSB	200 µL of oocysts (ca. 10 million) were mixed with 500 µL of 1 M MSB, vortexed, then held at room temperature for 4 h. The suspension was then washed 3 times with water before total RNA extraction
0.1 M MSB	200 µL of oocysts (ca. 10 million) were mixed with 500 µL of 0.1 M MSB, vortexed, then held at room temperature for 4 h. The suspension was then washed 3 times with water before total RNA extraction
Xanthine oxidase and hypoxanthine	200 µL of oocysts (ca. 10 million) were vortexed and 50 µL of xanthine oxidase (20 U/mL) added to the suspension. The suspension was brought up to 500 µL with PBS (pH 8.5) and incubated at 37 °C for 10 min. Thereafter, 500 µL of 0.5 mM hypoxanthine was added to the mixture, briefly vortexed and further incubated at 37 °C for 30 min with the lids open. The sample was then washed 3 times with water before total RNA extraction.
Heat shock	200 µL of oocysts (ca. 10 million) were incubated at 37 °C for 20 min. The sample was then washed 3 times with water before total RNA extraction.
Heat shock	200 µL of oocysts (ca. 10 million) were incubated at 45 °C for 20 min. The sample was then washed 3 times with water before total RNA extraction.
Control	200 µL of oocysts (ca. 10 million) stored at refrigeration temperature were washed five times with water before total RNA extraction.

**Table 2 microorganisms-09-01463-t002:** Oligos employed to evaluate the validity of selected target genes in the development of new RT-qPCR method to assess the viability of *Cryptosporidium* oocysts.

Target Gene	Forward Primer (5′ → 3′)	Reverse Primer (5′ → 3′)	Probe (5′ → 3′)	Product Size (bp)	Locus Tag (Ref.)
COWP7	CTATGGGATTCAATTTCGAAGTTCC	CCCAATACAAAATCTGCTACTTCCA	ATGGAATATCATCATCCCCTCAGCAA	97	cgd4_500
Thioredoxin	GAAAAGCTGAACCTCGCATTCG	CGTCCCGTGGTCAATGCAATAA	NA	134	cgd7_4080
Prohibitin	CCTTTTAGGTGCAATCGGAACA	CATGGGAGGAAGAAGTGGGTAC	NA	141	cgd7_4240
MDH	TCCTCTAGATGCGATGGTTTACTAC	CCACCTACAACAATGGCTGATACA	NA	162	cgd7_470
UGDH	CCTCCAACATTATCAGCTTTTTGAG	TGCATTTTAGAGTGAACCGCTT	NA	141	cgd8_920
HSP70	AGCCCGTATGAGTACAGAAGACT	GCCTGTGCCAAGAACCCTAAGA	NA	168	cgd4_3270
18S rRNA	JF1: AAGCTCGTAGTTggatTTCTGJF2: AAGCTCGTAGTTaatcTTCTG	JR: TAAGGTGCTGAAGGAGTAAGG	JT2: TCAGATACCGTCGTAGTCT	434	[18]

**Table 3 microorganisms-09-01463-t003:** Comparative quantitative analysis of selected target genes using 18S rRNA as the reference gene (normalizer).

Target Gene	Test Samples (C_q_) ^a^				Control Samples (C_q_)			Log2FC ^b^
	T1	T2	T3	Mean	C1	C2	Mean	
COWP7	28.7	28.9	28.3	28.6	34.6	35.1	34.8	6.34
Thioredoxin	23.4	23.5	23.2	23.3	27.3	27.3	27.3	4.08
UGDH	25.1	25.2	25.1	25.1	29.6	29.4	29.5	4.49
MDH	22.7	22.6	22.6	22.6	26.5	26.7	26.6	4.11
HSP70	24.6	24.0	24.5	24.3	28.9	28.6	28.7	4.54
Prohibitin	26.0	25.8	25.9	25.9	30.0	30.1	30.0	4.28

^a^ Samples subjected to oxidative stressor challenge using xanthine oxidase and xanthine reaction. ^b^ Log2fold change.

**Table 4 microorganisms-09-01463-t004:** Evaluation of the RT-qPCR targeting thioredoxin by using mixtures of viable and inactivated oocysts of *C. parvum*.

Sample Preparation			Induction Treatment Group		
	Viable (%)	Inactivated (%)	Control C_q_ Value	Oxidative Stress C_q_ Value	ΔC_q_
Sample 1	100	0	29.9	25.6	4.3
	100	0	30.0	25.8	4.2
Sample 2	10	90	34.9	29.6	5.3
	10	90	33.6	28.6	5
Sample 3	1	99	No C_q_	32.0	NA
	1	99	36.4	31.5	4.9
Sample 4	0	100	No C_q_	No Ct	NA
	0	100	35.9	36.0	−0.1

**Table 5 microorganisms-09-01463-t005:** The effect of extreme temperature on the viability of *Cryptosporidium* oocysts.

Treatment Condition		C_q_ Value COWP7		C_q_ Value Thioredoxin	
		No Oxidative Stress	Oxidative Stress	No Oxidative Stress	Oxidative Stress
−20 °C for 2 h	Replicate 1	34.1	28.4	26.9	24.6
	Replicate 2	34.3	28.5	26.7	23.6
	Replicate 3	33.7	28.4	26.9	23.9
	Mean C_q_ ± SD	34 ± 0.3	28.5 ± 0.1	26.8 ± 0.1	24.0 ± 0.5
−20 °C for 24 h	Replicate 1	33.7	35.3	30.2	30.2
	Replicate 2	33.9	34.7	30.4	30.0
	Replicate 3	33.6	33.7	30.2	29.1
	Mean C_q_ ± SD	33.7 ± 0.2	34.6 ± 0.8	30.2 ± 0.1	29.7 ± 0.6
−20 °C for 48 h	Replicate 1	36.6	37.8	30.7	30.8
	Replicate 2	37.9	37.6	30.7	32.8
	Replicate 3	35.1	36.4	29.9	31.2
	Mean C_q_ ± SD	36.5 ± 1.4	37.3 ± 0.7	30.4 ± 0.5	31.6 ± 1.1
60 °C for 2 min	Replicate 1	37.8	36.7	31.8	30.8
	Replicate 2	37.7	35.1	32.9	31.0
	Replicate 3	No C_q_	35.7	31.7	30.8
	Mean C_q_ ± SD	37.8 ± 0.1	35.8 ± 0.8	32.1 ± 0.7	30.9 ± 0.1
80 °C for 3 min	Replicate 1	No C_q_	No C_q_	No C_q_	No Cq
	Replicate 2	No C_q_	No C_q_	35.9	36.0
	Mean C_q_ ± SD	NA	NA	NA	NA
Control ^a^	Replicate 1	34.6	28.7	27.3	23.4
	Replicate 2	35.1	28.9	27.3	23.5
	Replicate 3	ND	28.3	ND	23.2
	Mean C_q_ ± SD	34.9 ± 0.4	28.6 ± 0.3	27.3	23.4 ± 0.2

^a^ Not exposed to the low/high temperature; ND Not done; NA Not applicable.

## Data Availability

The RNA-Seq data is available for public use at the sequence read archive of NCBI (http://www.ncbi.nlm.nih.gov/bioproject/669334, accessed on 7 July 2021).

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
