# Peer review of "Use of Oxidative Stress Responses to Determine the Efficacy of Inactivation Treatments on Cryptosporidium Oocysts"

_microorganisms, 2021, doi:10.3390/microorganisms9071463_

Round 1
Reviewer 1 Report
I am pleased to read the content of interesting results of your publication. Congratulations on the work, it's clearly written. Cryptosporidium spp., a leading cause of persistent diarrhea in developing countries, is a major threat to water and food supply worldwide. The ability to determine inactivation rates of Cryptosporidium oocysts in environmental samples is critical for assessing the public health hazard. The presented proof of concept (method) will certainly useful for risk assessment and of great practical importance.
Author Response
Thank you for taking the time to critically review our manuscript!
Reviewer 2 Report
This is an interesting, well prepared, and well-presented manuscript. The expertise and understanding of methodological details of this reviewer are not sufficient to evaluate or comment on the technical details of the presentation. The principal focus of this review has been on understanding and interpreting the substance of the procedures as reflected in comments included in the text.

Author Response
The authors would like to thank the reviewer for critically reviewing our manuscript and for the useful comment and suggestions. We have tried to reply to each comment and suggestion.

Reviewer 3 Report
Review of the MS No. microorganisms-1282863 entitled "Use of oxidative stress responses to determine the efficacy of inactivation treatments on Cryptosporidium oocysts"
The manuscript presents results of the study on assessment of the efficacy of Cryptosporidium oocysts inactivation. The study showed that RNA-Seq and DEG analysis are useful tools in the identification of targets, and RT-qPCR method can be alternative technique for testing the efficacy of various inactivation treatments of Cryptosporidium oocysts. The study is well designed, and the manuscript is well written. In my opinion the manuscript may be published in the present form.
Author Response
Thank you very much for taking the time to critically review our manuscript!
Reviewer 4 Report
This is a novel and interesting proof of concept paper that is very well described. Additional tables and figures in the supplemental material is very appropriate. I have only one query and a couple of very minor suggestions.
Line 90. Just to confirm that the gp60 subtype of the IOWA isolate purchased from Bunch Grass Farm is IIaA17G2R1 and this is not a typographical error. The IOWA II isolate has the gp60 subtype IIaA15G2R1 and while it has been shown that the genetic sequences of target genes of the IOWA isolate have changed slightly over time in different labs (Cama et al 2006, J. Eukaryot. Microbiol., 53(S1), 2006 pp. S40–S42), however, such a large change in the gp60 sequence has not reported to my knowledge.
Table 1. The spacing between treatments is inconsistent and it would simplify the table to add a row after each of the first 3 treatments, as was done with the last treatments.
Table 2.
-For consistency, insert 5' → 3' below the heading Probe in column 4
-Add (bp) below the heading Product size in column 5
-Add Locus tag in the heading in column 6, above [Ref]
Table 3.
Add Log2foldchange to the footnote of the table.
Author Response
The authors would like to thank the reviewer for critically reviewing our manuscript and for the useful comment and suggestions.
Regarding the question about the GP60 subtype. We have double checked our sequences and again confirmed that the Cryptosporidium isolate used in this study was in fact IIaA17G2R1. The isolate is similar to that of IOWA-ATCC, which is also the IOWA IIa type. According to the information obtained from CryptoDB, the GP60 locus of IOWA encodes 17 serines whereas IOWA-ATCC encodes 19 serines. https://cryptodb.org/cryptodb/app/record/genomic-sequence/CP044417#dnaContextUrl
We have revised the tables according to the reviewer's suggestion.